# Extreme transport of light in spheroids of tumor cells

**Davide Pierangeli** [1,2] ✉, **Giordano Perini**[3,4], **Valentina Palmieri**[1,3], **Ivana Grecco**[2], **Ginevra Friggeri**[3,4], **Marco De Spirito** [3,4], **Massimiliano Papi** [3,4] ✉, **Eugenio DelRe** [2] & **Claudio Conti** [2]

Extreme waves are intense and unexpected wavepackets ubiquitous in complex systems. In optics, these rogue waves are promising as robust and noise-resistant beams for probing and manipulating the underlying material. Localizing large optical power is crucial especially in biomedical systems, where, however, extremely intense beams have not yet been observed. We here discover that tumor-cell spheroids manifest optical rogue waves when illuminated by randomly modulated laser beams. The intensity of light transmitted through bio-printed three-dimensional tumor models follows a signature Weibull statistical distribution, where extreme events correspond to spatially-localized optical modes propagating within the cell network. Experiments varying the input beam power and size indicate that the rogue waves have a nonlinear origin. We show that these nonlinear optical filaments form high-transmission channels with enhanced transmission. They deliver large optical power through the tumor spheroid, and can be exploited to achieve a local temperature increase controlled by the input wave shape. Our findings shed light on optical propagation in biological aggregates and demonstrate how nonlinear extreme event formation allows light concentration in deep tissues, paving the way to using rogue waves in biomedical applications, such as light-activated therapies.

Many complex systems, including oceans[1], plasmas[2], lasers and optical media[3–8], are known to manifest rogue waves (RWs), extreme perturbations that are statistically rare in a fluctuating environment. Generally associated with unpredictable occurrences and catastrophes, RWs can also represent a route to transport and to localize energy in disordered systems. Their formation is supported by a variety of linear and nonlinear physical processes[9–13], which encompass chaos[14], turbulence[15,16], wave scattering[17,18] and topology[19], and whose interplay and unification is actively debated[20,21]. Their appearance is strongly dependent on the microscopic properties of the system, a fact that stimulates a wide-ranging research effort aimed at taming and

exploiting RWs in a variety of applications, including neuromorphic computing[22].

Biological systems are complexity-driven, forming a potentially rich and hereto unexplored field for optical extreme waves. In particular, living biomatter is an optically thick, agitated, absorbing, and highly scattering environment[23], features that make the concentration of visible light in deep tissues a critical problem[24]. In this context, RWs could be exploited as noise-resistant localized wavepackets that act as a source of intense light spots to probe, activate, and manipulate biochemical content. However, their formation in biological structures has not been reported yet.

[1]Institute for Complex Systems, National Research Council, Rome 00185, Italy. [2]Physics Department, Sapienza University of Rome, Rome 00185, Italy. [3]Neuroscience Department, University Cattolica del Sacro Cuore, Rome 00168, Italy. [4]IRCSS, Fondazione Policlinico Universitario Agostino Gemelli, Rome 00168, Italy. ✉e-mail: davide.pierangeli@roma1.infn.it; massimiliano.papi@unicatt.it

Here, we observe optical extreme waves that form in a dense spheroid of living tumor cells. The RWs propagate through the bioprinted three-dimensional tumor model (3DTM) for several millimeters, delivering extreme optical intensity and causing the transmitted light to have heavy-tailed statistics. The probability density of the intensity is found to be well modeled by a Weibull distribution with signature scale and shape parameters. We identify extreme events with spatially localized modes of enlarged transverse size. Importantly, the RWs exhibit complex dynamics when the input power is varied, which indicates the anomalous statistics originates from nonlinear effects. Measurements of the transmission matrix show these intense filaments constitute addressable channels with super transmittance. We analyze the RWs to achieve controllable large local temperature increases within the tumor spheroid by shaping the input excitation, thus enabling their possible application in photo-thermal therapy. These findings unveil a phenomenon able to transport massive optical power through strongly scattering biological matter.

## Results

### Observation of extreme waves in tumor spheroids

We investigate laser light propagation in millimeter-sized tumor spheroids fabricated via bioprinting from human pancreatic cells[25] (see Methods−Tumor spheroid growth). Tumor spheroids, namely 3DTMs, are widely used as cancer surrogates to study cell proliferation and drug response, and an increasing interest is currently emerging on their physics[26−29]. 3DTMs are formed by cells that are densely packed into an elastic network permeated by intercellular fluid. Unlike biological suspensions where cells are free to move under the action of light[30−32], tumor spheroids optically behave as a complex multiparticle assembly that strongly scatters visible radiation[33].

Coherent optical scattering from bioprinted tumors is largely unexplored. We study light transmission through pancreatic 3DTMs by using a spatially-modulated laser beam at $\lambda = 532$ nm. The input beam is richer in wavevectors than a collimated laser. We use a phase-only spatial light modulator (SLM) to shape randomly the wavefront of the optical field incoming on the tumor spheroid. Figure 1a illustrates the concept of our experiments. A bright microscopy image of a bioprinted 3DTM is shown in Fig. 1b along with the corresponding false-color map that shows the inhomogeneity of the cell network. The experimental setup, shown in Supplementary Fig. 1, is detailed in Methods. We observe light on the transmission plane forming a speckle pattern generated by multiple scattering. The measured diffuse reflectance is ≈0.3, which suggests that low-power light propagation is governed by diffusive transport[34]. At the operating wavelength, the measured absorption coefficient of the bioprinted spheroids without culture is ≈0.25 cm⁻¹, comparable with similar pancreatic tissues that are strongly scattering[35]. We hence expect the wave propagation to be described by a random Gaussian process for the field amplitude (random wave theory, RWT), which results in the Rayleigh statistics[36]. Depending on either the input beam power, size, and shape, two distinct scenarios are observed. At low input power ($P < 15$ mW), the speckle intensity distribution contains spots following RWT (Fig. 1c), hereafter referred to as normal waves (NWs). As we increase power, spatial modes with an anomalously large intensity appear (Fig. 1d). We will identify these bright peaks as RWs. To investigate the statistical properties of the transmitted field, we encode a set of random phase masks on the input wave, and collect the corresponding intensity distributions. Our method allows to propagate through the 3DTM the set of spectra $\mathbf{K} = \{\mathbf{k}_1,..,\mathbf{k}_N\}_i$, with $i = 1,...,L$, and each spectrum made of random wavevectors $\mathbf{k}_j$ that are uniformly distributed. The input spectrum is homogeneous, i.e., the field undergoes random interference when propagating in the absence of the biological sample.

Evidence of optical rogue waves in 3DTMs is reported in Fig. 2a. The probability density function (PDF) of the intensity presents a

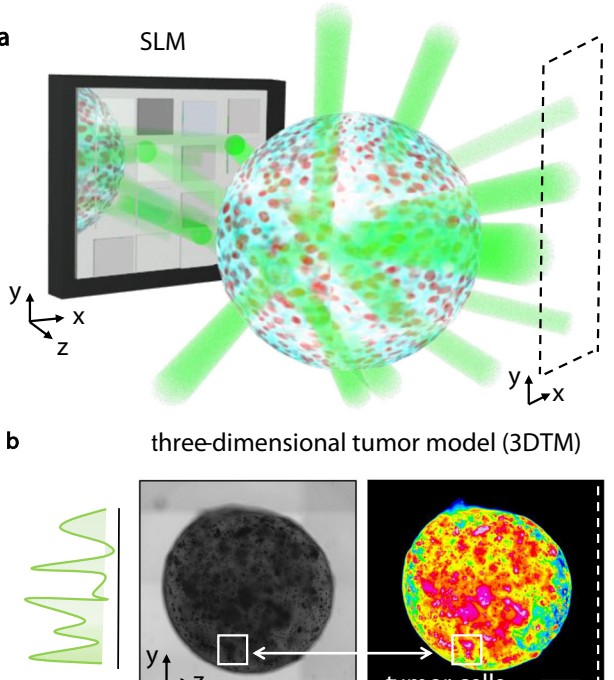

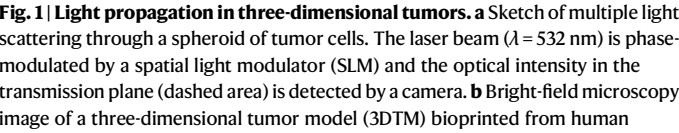

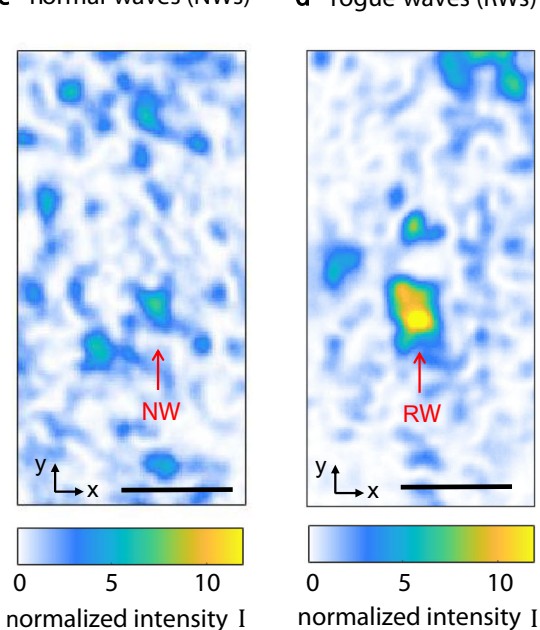

**Fig. 1 | Light propagation in three-dimensional tumors. a** Sketch of multiple light scattering through a spheroid of tumor cells. The laser beam ($\lambda = 532$ nm) is phase-modulated by a spatial light modulator (SLM) and the optical intensity in the transmission plane (dashed area) is detected by a camera. **b** Bright-field microscopy image of a three-dimensional tumor model (3DTM) bioprinted from human pancreatic cells. The false-color image highlights high-density regions. Scale bar is 500 μm. Speckle patterns showing the spatial intensity transmitted through the millimetric volume sample **c** at 5 mW input power (normal waves, NWs) and **d** in the presence of a rogue wave (RW) at 20 mW. Scale bar is 50 μm.

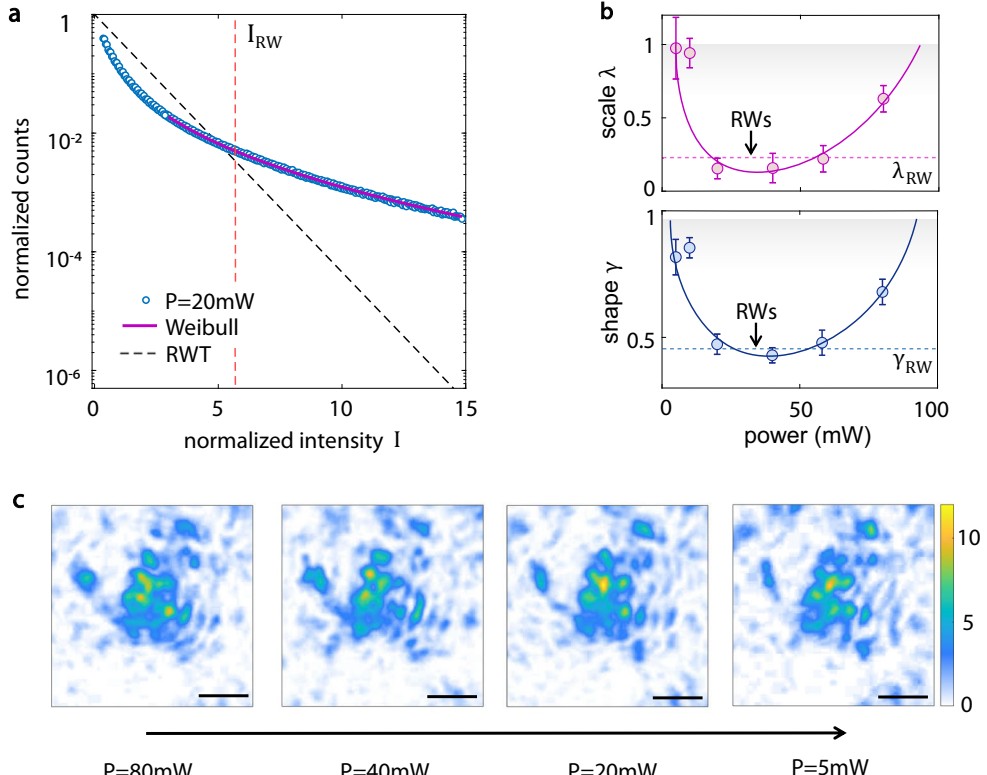

**Fig. 2 | Observation of optical extreme waves in tumor spheroids. a** Probability density of the transmitted intensity for input power $P = 20$ mW, showing the presence of rogue waves (RWs), i.e., a heavy-tail statistics that deviates from the Rayleigh distribution predicted by random wave theory (RWT). A Weibull distribution models the experimental data. $I_{RW}$ is the RW threshold according to the oceanographic criterion. **b** Shape $\gamma$ and scale $\lambda$ parameter of the Weibull function varying the input power. Strong RWs occur by increasing the input power, within an optimal power region. RWs are associated to specific parameters $\gamma_{RW}$, $\lambda_{RW}$. Lines are convex fit curves. The error bar is the standard deviation over 5 independent experiments. **c** Evolution of a RW as we decrease the input power, showing that the intense mode builds up and deforms when varying the input energy. Scale bar is 50 μm.

marked heavy tail, the signature of extreme wave occurrence. The heavy-tail statistics indicates that optical modes with extreme intensity form out of the 3DTM more frequently than expected from random wave theory (RWT)[9]. The observed PDF follows a Weibull distribution

$$f(I) = \frac{\gamma}{\lambda^{\gamma}} I^{\gamma-1} e^{-\left(\frac{I}{\lambda}\right)^{\gamma}}, \qquad (1)$$

where $\gamma > 0$ and $\lambda > 0$ are the shape and scale parameter, respectively. The measured Weibull parameters are shown in Fig. 2b as we vary the input laser energy (see "Methods−Experimental procedure"). Small deviations from RWT, which gives $\gamma = \lambda = 1$, are found at low power. Abundant RWs emerge in a broad power range that spans half an order of magnitude (20–70 mW). Exceeding this optimal power region, the statistics returns to approach RWT. At higher power ($P > 100$ mW), the output becomes unstable (non-stationary). Interestingly, when strong RWs occur, the scale and shape parameters have specific power-independent values, $\lambda_{RW} = 0.16 \pm 0.07$ and $\gamma_{RW} = 0.45 \pm 0.04$ (Fig. 2b). $\lambda_{RW}$ indicates a typical intensity fraction transported through the tumor-cell network.

To understand the RWs origin, we capture a RW generated at high power and observe its evolution as the laser energy is gradually decreased. The behavior, reported in Fig. 2c, reveals the extreme event as a self-interacting (nonlinear) wavepacket, i.e., a spatial optical mode that continuously changes its waveform when varying the input energy. In the shown case, the peak maximum features a non-monotonic behavior with respect to the total incoming energy, being enhanced for an intermediate power ($P = 20$ mW) that corresponds to the optimal power region observed in Fig. 2b. This power-dependent dynamics indicates that the RW formation is sustained by nonlinear

optical mechanisms[37]. Weakening and suppression of RWs at very high power is also a sign of their nonlinearity. In fact, in nonlinear disordered media, instability induces fast temporal dynamics and hence wave randomization[38]. The RW phenomenon is thus profoundly different from the long-tail statistics reported for biological tissues in optical coherence tomography[39] and diffuse reflectance[40], where the anomalous PDF comes from the complex shape of the biological scatterers and the multi-scale nature of the tissue structure. The observed phenomenological picture suggests the presence of optically-induced thermal effects and structural deformation of the cell network. It is known that cell nuclei regulate their shape and volume in a highly temperature-sensitive manner, and can exhibit volume transitions for laser-induced temperature variations of just a few degrees[41]. Observations in Fig. 2c show that the rearrangement of the tumor network induced by the input power is accompanied by the generation of RWs. We find that light-induced effects depend on the local 3DTM structure (Supplementary Fig. 2), in agreement with the presence of areas in which cells are more or less mobile and deformable[29].

The nonlinear nature of the extreme waves is confirmed by analyzing the spatial extremes of the wave field in analogy with studies on oceanic sea states[42]. We first identify each anomalous wave by using the oceanographic criterion extended to optical data[43], which defines the RW threshold as $I_{RW} > 2I_S$, where the significant intensity $I_S$ is the mean intensity of the highest third of events. Figure 3a reports the PDF of intensity maxima for the case in Fig. 2a, showing abundant peaks that significantly exceed $I_{RW} = 5.7$. We evaluate the Euler Characteristics (EC) for the amplitude field $\eta = \sqrt{I}$. The EC is a topological quantity that counts the number of connected components and holes within a field. Relevant is the EC of $U_{\eta h}$, with $U_{\eta h}$ the excursion set of $\eta$

of level $h$. In fact, for high threshold $h$, EC($U_{\eta h}$) gives the probability that a wave peak exceeds $h$[44]. The observed EC is compared with the EC expected for a linear (Gaussian) and nonlinear (Tayfun) wave surface (see Methods - Statistical analysis of the intensity field). The agreement with the nonlinear wave model indicates that nonlinear interaction intervenes in RW generation[45].

The emergence of RWs also depends on the spatial features of the input beam. Two opposite coupling conditions, referred to as weak and strong coupling, are obtained by varying the distance $d$ between the 3DTM and the input focal plane. Given an input field having a discrete wavevector spectrum $\mathbf{k}_j$ determined by phase modulation, via the optical geometry we alter the typical size of the modes impinging on the 3DTM. When the sample is placed after the focal region, modes of a wider transverse size excite the 3DTM. RWs appear in this condition (strong coupling, $d = 1$ mm), while they are suppressed in the weak coupling case ($d = 0$), with the PDF that closely follows RWT (see Supplementary Fig. 3).

We observe a notable difference in the size of intense waves for strong and weak coupling. RWs have a more irregular shape and their extension is $10 \pm 1\,\mu$m, larger than the mean speckle size ($7 \pm 1\,\mu$m). In Fig. 4a, b we compare the spatial positions of hundreds of intense events for experiments that give the Rayleigh and Weibull distribution. Normal waves (NWs) are distributed sparsely and quasi-homogeneously in the transmission plane (Fig. 3a), as expected from RWT. On the contrary, RWs are associated to specific spatial regions (Fig. 4b). We thus identify RWs in 3DTMs as localized optical filaments that form in specific regions. The large diameter of RWs allows us to understand the role of the coupling geometry. A focused input beam with small coherence length (weak coupling) is unlikely to match this size, whereas a more extended wavefront with wider modes can undergo self-focusing and form localized filaments. This size-dependent behavior is typical in optical soliton formation[46].

## Control of rogue waves for light delivery in tumors

To understand to what extent we can control RWs in 3DTMs, we measure the transmission matrix (TM) and analyze the transmission eigenchannels[47,48]. Generally, this approach is performed to study the transmission properties of well-controlled scattering systems, such as thin diffusers[49] and two-dimensional (2D) disordered waveguides[50], but still holds for nonlinear random media[51]. Here, importantly, we are applying the eigenchannels method to a macroscopic biophysical system.

The TM of a 3DTM at $P = 60$ mW is reported in Fig. S4 ("Methods−Transmission matrix"). We measure the matrix $t$, with complex coefficients $t_{nm}$, for $N = 64$ ($M = 900$) input (output) modes, for weak and strong coupling. We consider the transmission eigenvalues $\tau_n$ of the matrix $tt^\dagger$, which coincides with the squared values of the singular values of the TM, $t = U\sqrt{\tau}V^\dagger$. The eigenvalues give the transmittance of the optical eigenchannels supported by the cell network at a given input power, with the largest $\tau_n$ that correspond to high-transmission

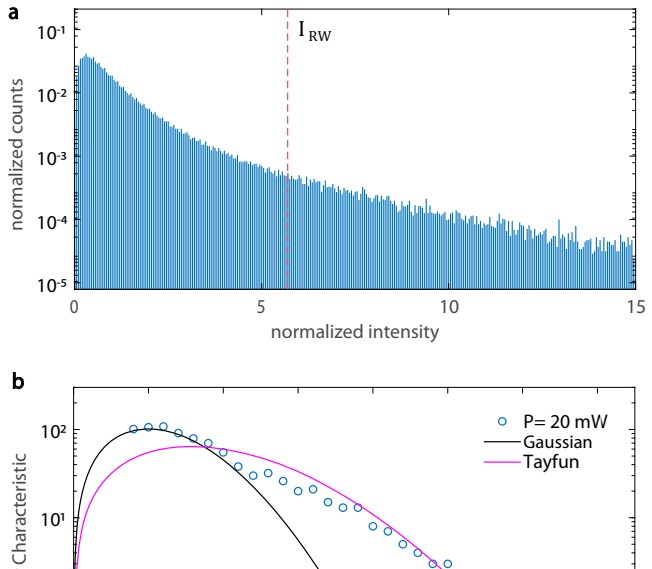

**Fig. 3 | Analysis of spatial extremes of the transmitted field. a** Probability density of the intensity maxima ($P = 20$ mW). Peaks exceeding the threshold $I_{RW}$ are classified as RWs (oceanographic criterion). **b** Euler Characteristics (EC) of the excursion set of level $h$ of the transmitted amplitude field, with the theoretical EC for a linear (Gaussian) and nonlinear (Tayfun) random wave field. $\sigma$ is the amplitude standard deviation.

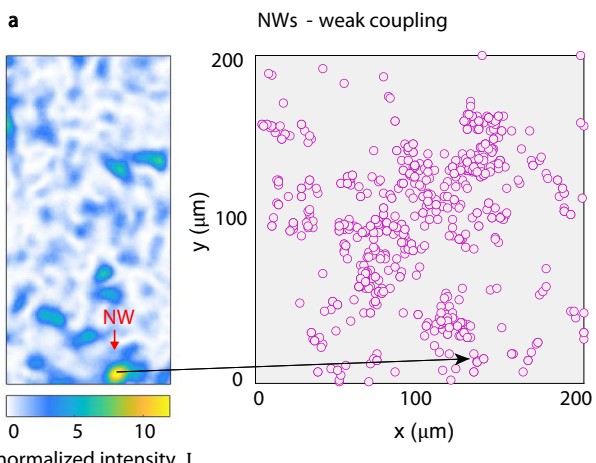

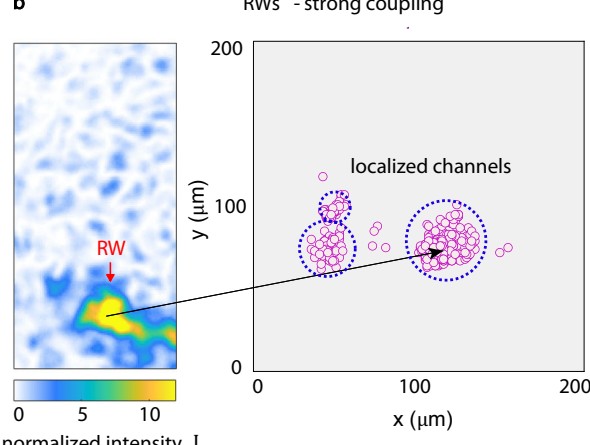

**Fig. 4 | Identification of localized waveguiding channels. a** NWs that follow RWT are sparsely distributed at random positions. Inset shows an example of a NW event. **b** RWs emerge in strong coupling experiments, i.e., for input modes of wider size. RWs have a large spatial extent and are all positioned into definite regions. These areas (dotted circles) indicate the position where the nonlinear waveguiding channels form.

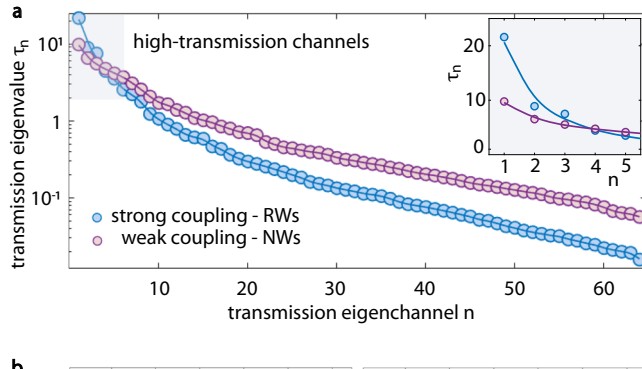

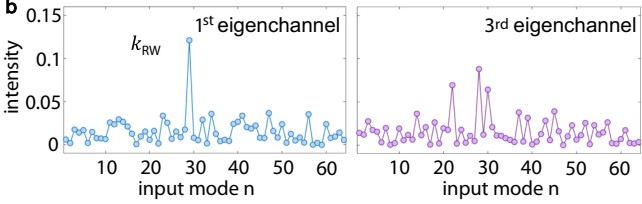

**Fig. 5 | Spectral analysis of the transmission channels. a** Measured ($P$ = 20 mW) transmission eigenvalues $\tau_n$ normalized and sorted in decreasing order, with the gray-shaded region that highlights high-transmission channels. The inset shows the eigenchannels with large transmittance for NWs and RWs. **b** Intensity profiles of the of the 1st and 3rd eingenchannel. The pronounced peak at $k_{RW}$ indicates the wavenumber that excites most RWs.

modes. The measured $\tau_n$ are reported in Fig. 5a in decreasing order. Surprisingly, we observe that, in the strong coupling condition that gives RWs, most channels have a reduced transmittance. Only a few channels transmit considerably more intensity than the average (Fig. 5a, inset). Therefore, RWs can be mainly associated to the first transmission eigenvalue. The intensity profile of the 1st eingenchannel is shown in Fig. 5b. The sharp peak at a specific input mode indicates the input wavenumber $k_{RW}$ exciting RWs. Similar peaks are found for the 2nd and 3rd eingenchannel (Fig. 5b). Interestingly, transmission channels with similar localized spectra have been predicted to support intense optical filaments[52].

The ability to deliver laser beams deep within tumors is essential for phototherapy, which is currently hampered by the limited penetration depth of visible light[53]. Optical waveguiding through thick tumors is here demonstrated via the spontaneous formation of RWs, which behave as waveguided modes carrying extreme intensity on a micrometric spot. Large optical power on a given target induces local temperature variations that can alter the functionalities of specific cells or to activate drugs and photosynthesizers. Hereafter, we illustrate the application of extreme nonlinear waves for photo-thermal therapy. We use the transmitted intensity in a setup where light-to-heat conversion occurs by nanoparticles in aqueous solution, a scheme common in biomedical applications[25]. Following the model in Ref. 54, we evaluate the local temperature increase induced upon the area where huge RWs ($I > 2I_{RW}$) emerge ("Methods−RW-induced temperature increase"). Figure 6a shows the temperature change $\Delta T$ produced by each randomly modulated input. The most intense RWs gives temperature spikes exceeding 10 °C. The input phase mask that generates an extreme wave (Fig. 6b) produces a local temperature enhancement significantly larger than achievable with an homogeneous or random input. This suggests the prospect of effective treatment of deep tissues by tailored beams and nonlinear propagation.

## Discussion
The observation of extreme waves in 3DTMs raises the question if they can have different features in healthy spheroids. More generally, it opens the discussion on the way the formation of RWs depends on the cell type and network structure. In this direction, we perform an

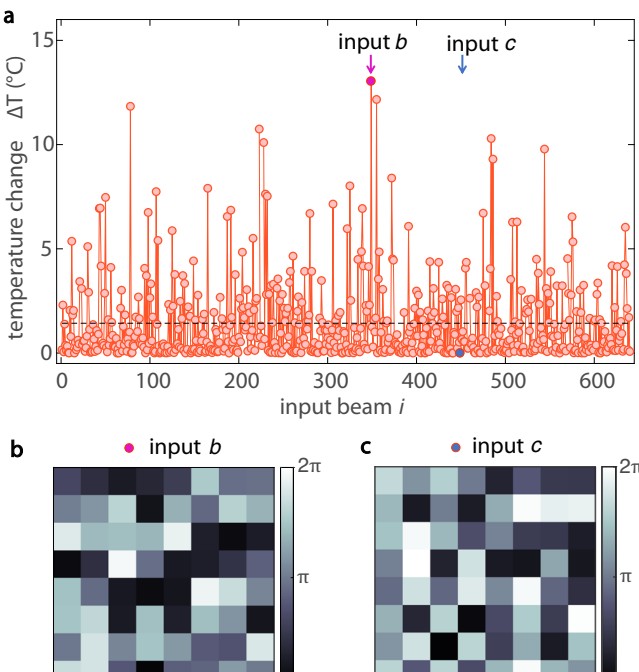

**Fig. 6 | Local temperature increase induced by extreme waves. a** Temperature change within a RW channel for different randomly modulated input beams. Light-to-heat conversion occurs via dispersed nanoparticles. Large temperature increases with respect to the average (dotted line) occurs when RWs are excited. **b, c** Input phase masks giving the maximum and negligible $\Delta T$.

experimental investigation on our 3DTMs treated with chemotherapy (Gemcitabine, 100 μM), which inhibits cell replication. We observe no systematic differences in the intensity statistics and RW properties. This suggest that RWs in tumor cells are robust to the presence of drugs.

To conclude, we have observed that macroscopic tumor spheroids, bioprinted from human pancreatic cells, support optical extreme waves. This demonstrates nonlinear optical rogue events in biophysical structures, opening a new scenario for the application of extreme waves. Our observations reveal that a Weibull distribution characterizes laser transmission in tumor models, and demonstrate intense, localized, naturally-occurring light transport over millimeters through dense cell aggregates. These findings may introduce new optical approaches for biomedical applications. Among these, the exciting possibility of treating cancer using tsunamis of light.

## Methods
### Experimental setup
A picture of the experimental setup is reported in Supplementary Fig. 1. A CW laser beam with wavelength $\lambda$ = 532 nm and tunable output power up to 2W (LaserQuantum Ventus 532), linearly polarized along the experimental plane (x-axis), is expanded and impinges on a reflective phase-only SLM (Hamamatsu X13138, 1280 × 1024 pixels, 60 Hz). The SLM active area is divided into $N$ = 64 squared input modes by grouping 120 × 120 pixels, with each mode having a phase $\phi_j$. The available phase levels are 210, linearly distributed within [0, 2π]. The phase-modulated beam is spatially filtered and focused by a plano-convex lens ($f$ = 100 mm, NA = 0.4) into a 3DTM sample positioned along the beam path (z-axis) utilizing a sample holder equipped with a three-axis translational stage. The input power $P$ is measured close before the sample. The 3DTM is immersed in its fresh culture solution in a 10 mm long optical-quality quartz cuvette. The average diameter of the 3DTM samples is 3 ± 0.2 mm. The spheroid position $d$ with respect to the input focal plane is varied in the 0.5−5 mm range to

change the optical coupling. The transmitted intensity on the cuvette output facet (transmission plane) is imaged by an objective lens ($f = 75$mm, NA = 0.5) on a CMOS camera (Basler a2A1920-160umPR) with 12-bit sensitivity (4096 gray-levels). The entire setup is enclosed in a custom-built incubator kept at the constant temperature of 35 °C and under constant 5° $CO_2$ influx.

## Experimental procedure

For fixed experimental conditions (3DTM sample, input power, coupling geometry, etc.), we collect $L = 640$ transmitted intensity distributions, each for a different randomly-modulated input wave. We refer to this dataset as the outcome of a single experiment. Data are obtained sequentially at 10 Hz sampling rate by loading on the SLM the $L$ phase masks. Every mask is made by blocks with random phases $\phi_j$ drawn from a uniform distribution. Within the measured image, we select a $200 \times 200$ $\mu m^2$ region of interest (ROI), a size that ensures we analyze a speckle pattern that is homogeneous on a large scale. To position the ROI, we adopt the average position of the intensity center of mass (CM) as the origin of the $xy$-plane. Variations of the CM for a set of input phase masks are shown in Supplementary Fig. 6.

## Statistics and reproducibility

We carry out three replicated experiments (statistical replicates) for a given setting, showing similar results. Then we change the input power, and the coupling geometry, but keeping the same 3DTM sample. The whole set of experiments is repeated over five different 3DTM samples (biological replicates). This measurement campaign is performed within a day. It has been repeated for five days within a week, each day using a new set of 3DTM samples. No qualitative differences are found between the results of different days. Microscopy of different 3DTM samples shows similar features. Parallel experiments have been performed on a line of 3DTMs treated with chemotherapy (Gemcitabine 100 μm).

## Statistical analysis of the intensity field

In oceanography, RWs are defined as waves whose trough-to-crest height exceeds twice the significant wave height, the mean height of the highest third of waves. In optics, the definition becomes $I_{RW} > 2I_S$, with $I_S$ the mean intensity of the highest third of events. Measured intensity data are normalized to the mean intensity $\langle I \rangle$. Data (matrices of size $200 \times 200$) are processed to find local maxima, which form the series of events used to evaluate $I_S$ for that set of experimental parameters. The PDF in Fig. 2a is measured on a single tumor spheroid. The dataset has 25,600,000 points and the values are shifted by $\langle I \rangle$ to remove the incoherent component (background). In this case we count approximately 2700 RWs (Fig. 3a). For the EC analysis, the excursion set $U_{\eta h}$ is computed as the portion of the image where the wave amplitude $\eta = \sqrt{I} > h$. The Euler number is $EC(U_{\eta h}) = CC - H$, being $CC$ the number of connected components and $H$ the holes. It is computed on the binary matrices $U_{\eta h}$ in MATLAB. The expected EC for a random wave field (Gaussian) is $EC_G = N_S \xi_G \exp(-\xi_G^2/2)$, being $\xi_G = h/\sigma$, with $\sigma$ the standard deviation of $\eta$, and $N_S$ the number of waves on the area. The nonlinear EC (Tayfun) is $EC_T = N_S \xi_T \exp(-\xi_T^2/2)$, where $\xi_T = (-1 + \sqrt{-2\mu\xi_G})/\mu$ with wave steepness $\mu$. For further details see ref. 45. The maps in Fig. 4 are obtained considering the spatial coordinates of events exceeding an intensity $I^* = 10$.

## Transmission matrix

The TM models monochromatic transmission through an optical system in terms of input and output modes. Its complex-valued entries $t_{nm}$ connect the amplitude and phase of the optical field between the $m$-th output and the $n$-th input mode, $E_m^{out} = \sum_{n=1}^{N} t_{nm} E_n^{in}$. To measure the TM of a 3DTM within a single experiment, we employ the $L = 10 \times N = 640$ random phase masks as independent realizations of the input field on $N$ input modes, and we select $M = 900$ output modes within the

camera ROI. Output modes are obtained by binning over a few camera pixels, and they have a size comparable with the spatial extent of a speckle grain ($7 \pm 1$ μm). The TM is reconstructed from the entire set of intensity data using a phase retrieval algorithm[55]. When we vary only the coupling conditions (Supplementary Fig. 4), the TM for strong and weak coupling are found to be correlated. This indicates that many of the scattering paths do not vary between the two measurements. The $\tau_n$ are normalized to the mean transmission $\langle \tau_n \rangle$. According to RWT, the $\tau_n$ probability density should follow a precise scaling known as Marcenko–Pastur law[47]. We found a strong deviation from this behavior (see Supplementary Fig. 5).

## Tumor spheroids growth

Three-dimensional tumor models (3DTMs) have been prepared from human pancreatic cells. The cancer cell line PANC-1 was purchased from the American Type Culture Collection (ATCC). Cells were maintained in Dulbecco's modified Eagle's medium (Sigma-Aldrich) supplemented with 10% fetal bovine serum (FBS, EuroClone), 2% penicillin-streptomycin and 2% L-glutamine (Sigma-Aldrich). Cells were cultivated in T75 flasks and kept at 37 °C, 5% $CO_2$. Cancer spheroids were produced via bioprinting. As a bioprinting strategy, droplet print of spheroids was used (BIOX, Cellink). For this purpose, $3 \times 10^6$ cells/mL were mixed with alginate (Sigma-Aldrich) at 5% w/v on a syringe with a 1:1 ratio. Cells mixed with the hydrogel were loaded on a bioprinting cartridge. Spheroid droplets were released on a 96-well round bottom (Corning) previously filled with 100 μL of 2% $CaCl_2$ to induce crosslink of alginate. Droplets were incubated for 5 min at room temperature (25 °C), then $CaCl_2$ was replaced with fresh complete culture medium. Spheroid droplets were then incubated at 37 °C, 5% $CO_2$ for further treatments.

## Monitoring of bioprinted 3DTMs

We monitored the growth of the bioprinted spheroids by a specifically-developed microscopy analysis tool[56]. Optical and fluorescence microscopy were carried out by using Cytation 3 Cell Imaging Multi-Mode Reader (BioTek). We collected bright-field data in a time span of 30 days and processed them via the Fiji software. The growth of spheroids was monitored over time in terms of radius and cell density. Representative confocal microscopy images were obtained with an inverted microscope (Nikon A1 MP+, Nikon). For these measurements, bioprinted 3DTMs were stained with Calcein-AM at a final concentration of 10 μM, and incubated at 37° C, 5% $CO_2$ for 15 min. Microscopy results are reported in Supplementary Fig. 7. The optical transmission experiments are performed two weeks after the spheroid growth. This guarantee that the bioprinted 3DTMs are stable over time (Supplementary Fig. 8).

## RW-induced temperature increase

We consider a setup where our 3DTM is immersed in water containing a colloidal suspension of 20 nm Au nanoparticles (NPs) and laser illuminated. NPs absorb light and generate heat dissipated into the environment, giving a macroscopic temperature increase. The effect is analyzed via a thermal transfer model that neglects convection and assumes uniform background heat conductivity[54]. This corresponds to a typical biomedical situation where hot NPs are injected into a small cavity inside a massive tissue and optically pumped. When the pumping beam is modeled as a uniform optical filament with spot area $A_b$, radius $R_b$, optical lenght $l_b$, and total optical intensity $I_0$ (W/cm²), the local temperature change produced by the laser beam is

$$\Delta T = \Delta T_{max} R_{NP} n_{NP} A_b \times 2 \ln(l_b/R_b), \quad (2)$$

where $n_{NP}$ is the average density of NPs of radius $R_{NP}$. $\Delta T_{max}(I_0) = c_w I_0$ is the temperature increase in proximity of the surface of the NP, being

$c_w$ a constant that depends on NP plasmonic response and water thermal conductivity[54]. To evaluate Eq. (2) by using the measured set of intensity patterns, we select as a spot area the spatial region where at least a RW with $I > 2I_{RW}$ is detected over the entire set. For each input beam in our experiment, the total intensity transmitted within this area is used as $I_0$. Therefore, the RW is modeled as a optical filament with the measured transverse intensity profile and optical length equal to the 3DTM thickness. We consider that nearly 1mW of total power falls within the selected ROI. The results in Fig. 6a are obtained using the values in Ref. 54 for the NP parameters.

## Reporting summary

Further information on research design is available in the Nature Portfolio Reporting Summary linked to this article.

## Data availability

All relevant data that support the findings of this study are available within the article and Supplementary information. Any additional data are available from the corresponding author upon request.

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

## Acknowledgements

We acknowledge funding from the Italian Ministry of Education, University and Research PRIN 2017 n. 20177PSCKT and AIRC IG 2019 n. 23124. We thank I. MD Deen and A. Augello for technical support in the laboratory, and S. Sennato for help in managing the biological samples. We thank the 3D Bioprinting Research Core Facilty G-STeP.

## Author contributions

D.P., V.P., M.P. and C.C. initiated the research line. D.P. realized the experimental setup and developed the experimental method. D.P. and I.G. carried out the experiments. G.P., V.P., G.F., M.D.S. and M.P. fabricate and characterize the biological samples. D.P. performed data analysis. D.P., E.D.R. and C.C. elaborated the interpretation of results. D.P, E.D.R and C.C. wrote the paper with contributions from all the authors.

## Competing interests

The authors declare no competing interests.
