## [Peer review file · Nature Communications]

Reviewers' comments:

Reviewer #1 (Remarks to the Author):

In the present manuscript the authors discuss the formation of optical rogue waves in a dense spheroid of living tumor cells. They launch a laser beam through the spheroids and the optical intensity in the transmission plane is detected by a camera; a speckle pattern arising from coherent multiple scattering is observed. The distribution of the intensity displays spots with an anomalously large intensity. The probability density of the intensity presents a heavy tail, the signature of rogue waves occurrence. A two parameters Weibull distribution is used to fit the PDF. The process of formation of rogue waves is linear as it seems to be independent of the intensity of the input beam.

The manuscript is in general well written (although it does not seem to be in the standard Nature-like format). In my opinion the results are interesting and original and could deserve publication in Nature Communications; however, I'm still confused on what is the main result reported. If I'm not wrong, the mechanism of formation of optical rogue waves as the light passes through a nonhomogeneous medium is not new: the group of Residori has already published some relevant work on the subject. If the authors agree on this, then I would like to know explicitly what their message in the field of biophysics is. What I do not understand is if the discussed property of formation of rogue waves is a peculiarity of spheroids tumor cells or rogue waves would also form from a healthy tissue. The authors conclude their manuscript with the following sentence: "... Among these, the exciting possibility of treating cancer using tsunamis of light". I fully understand the enthusiasm of the authors; however, have the authors made any quantitative calculation on the intensity of light needed for treating tumoral cells?

To conclude, what a biophysicist would learn from reading the present paper?

I have also some the following minor point: what are the statistical properties of the input beam?

Reviewer #2 (Remarks to the Author):

The paper deals with the observation and investigation of optical rogue waves in a biological system. Millimeter-sized tumor spheroids illuminated by randomly modulated laser beams are shown to lead to a transmission of light intensities with Weibull distribution. The deviation from statistical behavior following the random wave theory is interpreted as a signature of rogue waves. By measuring the transmission matrix, the authors demonstrate that the rogue events are related to spatially localized transmission modes. Structural deformation of the cell network and optically induced thermal effects are identified as the mechanism of generation.

I have several concerns about some main aspects of the investigation. Especially, I don't see that the data provided suffices to support the authors' claims. Some arguments appear unclear and contradictory. Therefore, I cannot recommend this manuscript for publication in the present state.

The authors' main claim is to observe optical rogue waves in a biological system for the first time. Their interpretation builds on the observed deviation of the statistical appearance of transmitted light with anomalously high intensities from statistics one would expect from the linear random wave theory. Biological tissues are known to provide complex scattering behavior which defy Gaussian statistics. A number of different probability density functions have been taken into account by other researchers to describe, e.g., speckle statistics of biological tissues in optical coherence tomography, see e.g. [Ge, Rolland, Parker, Biomedical Optics Express 12, 4179 (2011)]. Also the Weibull distribution has been investigated in connection with light scattering by biological tissues, e.g. [Fawzym Zeng, Applied Optics 45, 3902 (2006)]. There is a huge literature concerning non-Rayleigh statistics in biological tissues or turbid media, but there is no mention of any of it in the manuscript

and the work is not set in an appropriate context.

Usually, rogue waves are defined by a certain amplitude exceeding a significant wave height of the system. There is no corresponding value mentioned in the manuscript. The authors only refer to extreme or anomalous high intensities, but they draw a difference between bright spots arising from random Gaussian statistics and spots observed in the heavy tail statistics. A much clearer definition and justification of rogue waves in the systems would be helpful. From the point view of statistical appearance, bright spots with the same extreme and anomalous high intensities can also appear in the case of Gaussian statistics, which are even more rare but should still be observable. But those spots would be of a different type and would also exhibit completely different characteristics, as one has to conclude from Fig.3. Obviously, a transmission of light with high intensities is possible in different ways. This raises a lot of questions. For example, what is the other physical mechanism for the transmission? Is only one type observed in the two different scenarios?

Why do the authors need the concept of rogue waves at all? The authors basically demonstrate that the system can easily be controlled, in the statistics as well as in a direct control of the event itself. This appears to be in contrast with the unpredictability criterion of the rogue wave phenomenon. Especially, I don't see any gain in deeper insight concerning the rogue wave phenomenon.

How has the spatial intensity distribution for normal waves in Fig. 1 been received? Does it correspond to one realization observed in the investigation or has it been generated as a special case? This question concerns also the data which has been taken into account for the probability density function presented in Fig. 2 a. Does it contain the intensities of all measurements for one sample, or only for one random input wavevector spectrum $\{k_j\}$? This question is also related to my question above concerning the two scenarios.

The authors argue that the rogue wave formation is dominated by linear wave phenomena. It is difficult to follow this line of thought based on the data in Fig.2c. There are different numbers of peaks with high intensities for different powers and a not clear distribution to the other eigenchannels. The dynamics appear more complex, and the conclusion drawn from one peak for the total incoming energy is not directly accessible. The identification as a linear theory is an extremely important point, which should be supported by more clear data.

The difference of spatial properties for weak and strong coupling is interpreted as resulting from a reduced spectral content for the weak coupling. Again, this is not obvious by the provided data. How has this been investigated? The authors emphasize on the light-induced thermal effects. This adds a further parameter into the coupling principle. Can the authors exclude the generation of rogue waves for other input powers for the weak coupling case?

The authors state that rogue waves in photonics form a noise-resistant tool for probing and manipulating the underlying material. This point is not clear. The authors should support this statement with a corresponding reference.

Response letter to Reviewers

We thank the Reviewers for the valuable analysis of our work. Considering the scientific points emerged during the peer-review, we have performed further studies and profoundly revised our manuscript considering the new results we obtain. The paper has been rewritten around two key points.

First, the nonlinear origin of the extreme waves. The phenomenon we report is radically different from non-Rayleigh speckles in inhomogeneous media and biological tissues. The observed spots of extreme intensity are nonlinear modes, as we now demonstrate through a topological analysis of the wave field. The new results in Fig. 3, and the observations in Fig. 2 and Fig. 4, strongly support the picture in which nonlinear light filaments form and propagate through the tumor spheroid.

Second, the significance of our findings for biomedical applications. We show in the new Fig. 6 that rogue waves in tumors have direct application in photo-thermal therapy where they enable a large temperature increase in a localized spatial region.

We have addressed and clarified all the raised points. They are discussed in detail in the response below with direct links to the new paper.

Point-by-point response to Reviewer #1

Report 1: In the present manuscript the authors discuss the formation of optical rogue waves in a dense spheroid of living tumor cells. They launch a laser beam through the spheroids and the optical intensity in the transmission plane is detected by a camera; a speckle pattern arising from coherent multiple scattering is observed. The distribution of the intensity displays spots with an anomalously large intensity. The probability density of the intensity presents a heavy tail, the signature of rogue waves occurrence. A two parameters Weibull distribution is used to fit the PDF. The process of formation of rogue waves is linear as it seems to be independent of the intensity of the input beam.

The manuscript is in general well written (although it does not seem to be in the standard Nature-like format). In my opinion the results are interesting and original and could deserve publication in Nature Communications; however, I'm still confused on what is the main result reported.

Response: We thank the Reviewer for the positive report and very useful comments. We are pleased that they find our work interesting and original. Considering the Reviewer's comments, we realized that in the original paper we failed to convey some of the key points of our findings. Specifically, we did not make clear that the formation of optical modes with extreme intensity in tumor spheroids (i) is nonlinear in origin and (ii) has important implications for biomedical applications. We thus largely revise the paper, adding additional data and analysis that (i) prove that nonlinear effects underlie the rogue waves and (ii) show how these intense modes can be exploited in photo-thermal therapies.

Stated clearly, our main result is the observation of nonlinear optical modes that allows the transport of intense localized light through macroscopic cancer tissues.

Comment 1.1: If I'm not wrong, the mechanism of formation of optical rogue waves as the light passes through a nonhomogeneous medium is not new: the group of Residori has already published some relevant work on the subject. If the authors agree on this,

Response: No, in our case the RW formation mechanism is not the wave interference effect studied by Residori et al. in nonhomogeneous media (Ref. 11), where the long-tail statistics is produced by large-scale inhomogeneity of the field. Unfortunately, in the original version, the way we mixed linear and nonlinear concepts for explaining our observations resulted in a confusing message. To be clear, the RWs originate from nonlinear interaction, meaning that their physics cannot be described with linear optics.

The signature of the RW nonlinear nature is the dependence on power, which is absent in any purely linear wave scenario. We limit our study to 100 mW as non-reproducible effects occur at

larger powers. However, note that the suppression of RWs at very high power is also a sign of their nonlinearity. In fact, in nonlinear disordered media, nonlinear instability induces fast temporal dynamics and hence wave randomization (see Ref. 37). The behavior observed varying the input power (Fig. 2c and Fig. S2) suggests that optomechanical interactions, presumably mediated by thermal effects, have an important role.

In order to prove the presence of nonlinear effects also beyond the dependence on power, we estimate the Euler Characteristics (EC) of the measured 2D amplitude field, finding interesting results. In particular, following *Fedele et al, Mathematics and Computers in Simulation 82, 1102, 2012*, we compare the calculated EC with the Gaussian EC and the nonlinear Tayfun EC. The results, also shown in Figure R1 below (details are in the main article and methods), indicate that nonlinear effects influence the shape of the most intense waves in the transmitted disordered field.

Therefore, the formation of local nonlinear structures is the appropriate interpretation of our observations. Please see also the detailed discussion in response to Comment 2.1 of Reviewer #2 about the marked difference between our spatial RWs and non-Gaussian speckles.

Figure R1. (a) Probability density of the intensity maxima ($P=20$ mW). Peaks exceeding the threshold I_{RW} are classified as RWs. (b) Euler Characteristics (EC) of the excursion set of level h for the measured amplitude distribution, with the theoretical EC for a linear (Gaussian) and nonlinear (Tayfun) random wave field.

Revision: The interpretation of the results has been clarified throughout the paper, also correcting imprecise statements and making more clear the presentation of results in Fig. 2b [change (1)]. Figure 4b now reports the EC analysis [change (2)], and discussions on this method have been added throughout the main text with relevant references [change (3)].

Comment 1.2: I would like to know explicitly what their message in the field of biophysics is.

Response: Nonlinear optical waves for light transport in deep tumor tissues, this is the crucial message. In fact, light-based therapeutic methods (phototherapy) have recently emerged as a promising solution for cancer treatment due to their multifunctionality and minimal invasiveness but face the critical limitation of the visible light penetration depth that is restricted to a few millimeters (see, for example, *Lee et al, Advanced Drug Delivery Reviews 186, 114339, 2022*). The ability to form intense localized beams through millimeter-sized tumors on a given target is crucial to induce local temperature variations that can alter the functionalities of specific cells (photo-thermal therapy), or to locally activate chemical content and excite photosynthesizers. We demonstrate that light concentration occurs spontaneously by the formation of spatial rogue waves, which behave as waveguided modes delivering extreme intensity on a micrometric spot through the tumor spheroid.

To illustrate the potential of the observed RWs for phototherapy, we consider their use in a possible setup where light-to-heat conversion occurs by nanoparticles in aqueous solution, a scheme common in biomedical applications. Following *Richardson et al, Nano Letters 9, 1139, 2009*, we evaluate the local temperature change induced by the transmitted intensity in the area where RWs emerge. The model and the analysis are detailed in methods. As shown by the results in Figure R2 below (new Fig. 6 of the manuscript), which refer to the experimental data giving the long-tail statistics of Fig. 2a, shaped input beams that excite a RW (input *b*, for example) give a significant temperature increase, much larger than homogeneous or other randomly modulated beams (input *c*). This suggests that visible-light manipulation and treatment of deep tissues can be achieved by specific input modes.

Revision: To underline the importance of our work for phototherapy, we revise the abstract and introduction, and add a paragraph discussing the application of RWs for photo-thermal therapy [change (4)]. The expected temperature change induced by output speckles within a given spatial region is reported in Fig. 6 [change (5)]. Details on the light-to-temperature conversion analysis are reported in a new method section [change (6)].

Comment 1.3: What I do not understand is if the discussed property of formation of rogue waves is a peculiarity of spheroids tumor cells or rogue waves would also form from a healthy tissue.

Response: This is an extremely interesting point. The way the formation of optical RWs in cell aggregates depends on the cell type and structural properties of the network is an entire topic opened by the observations here reported. We have performed a first investigation in this direction. We compare our results in naturally evolving 3DTMs with the same spheroids treated with chemotherapy (Gemcitabine, 100 μ M) that inhibits cell replications, but we did not find systematic differences in the statistics of the transmitted intensity nor in the features of the RWs. This evidence is interpreted by considering that, for dense spheroids, the RW formation is found as independent of the cell density (as indicated by experimental results obtained after a

Figure R2. (a) Local temperature change in regions supporting extreme intensity modes ($I > 2I_{RW}$) as calculated from the measured intensity distributions. Temperature variations are obtained using the model and parameters in *Richardson et al, Nano Letters 9, 1139, 2009*, for heat conversion mediated by gold nanoparticles in water. The black line indicates the mean temperature change expected on the selected area when averaging over all the inputs. (b-c) Phase patterns of the input beam that give (b) the maximum $\Delta T = 13^\circ\text{C}$ and (c) no RWs and negligible temperature increase.

week, see Fig. S8). Research on 3DTMs made with different cells is ongoing but goes beyond present scopes.

Revision: We discuss the point, mentioning results in 3DTMs with chemotherapy [change (7)].

Comment 1.4: The authors conclude their manuscript with the following sentence: "... Among these, the exciting possibility of treating cancer using tsunamis of light". I fully understand the enthusiasm of the authors; however, have the authors made any quantitative calculation on the intensity of light needed for treating tumoral cells?

Response: Yes, we have followed the Reviewer suggestion. As per Comment 1.2, we have performed a quantitative calculation that links the measured extreme intensity to the heat that is locally generated by dissipation via a biophysical solution. The estimated local temperature variation exceeds 10°C for extreme waves, which is sufficient for an effective treatment. In fact,

heating of tissues to a temperature of 42–46 °C for a few minutes generally results in cell necrosis. Considering our additional analysis, we believe that our enthusiasm is now much more supported by the evidence.

Comment 1.5: To conclude, what a biophysicist would learn from reading the present paper?

Response: In a few words, they will learn that light can spontaneously propagate and concentrate deep into biological structures by forming nonlinear filaments of extreme intensity. Considering that localizing an intense laser on a micrometer spot through a millimetric tissue is currently a challenge, we believe the observed phenomenon can significantly impacts bio-sensing and light-based therapies.

Comment 1.6: I have also some the following minor point: what are the statistical properties of the input beam?

Response: The phases on the SLM modes are drawn randomly from a uniform distribution, i.e., the input beam is composed of a random set of wavevectors. Linear free-space propagation thus results in a Gaussian-distributed far-field intensity. We now specify this point [change (8)].

Point-by-point response to Reviewer #2

Report 2: The paper deals with the observation and investigation of optical rogue waves in a biological system. Millimeter-sized tumor spheroids illuminated by randomly modulated laser beams are shown to lead to a transmission of light intensities with Weibull distribution. The deviation from statistical behavior following the random wave theory is interpreted as a signature of rogue waves. By measuring the transmission matrix, the authors demonstrate that the rogue events are related to spatially localized transmission modes. Structural deformation of the cell network and optically induced thermal effects are identified as the mechanism of generation.

I have several concerns about some main aspects of the investigation. Especially, I don't see that the data provided suffices to support the authors' claims. Some arguments appear unclear and contradictory. Therefore, I cannot recommend this manuscript for publication in the present state.

Response: We thank the Reviewer for the profound analysis of our manuscript that helped us to arrive at a new paper that is considerably improved. We seriously addressed the Reviewer's main concern by performing an extensive analysis of the wave field and its dependence on power. This in-depth study allows us to fully understand the experimental observations and their applicative potential. In particular, we clarify that (i) extreme waves in tumor spheroids are nonlinear modes and (ii) can be controlled via the input beam as high-transmission channels. Our claims are now precise and strongly supported by the experimental evidence, so we are hopeful that the Reviewer will find the revised paper convincing.

Comment 2.1: The authors' main claim is to observe optical rogue waves in a biological system for the first time. Their interpretation builds on the observed deviation of the statistical appearance of transmitted light with anomalously high intensities from statistics one would expect from the linear random wave theory. Biological tissues are known to provide complex scattering behavior which defy Gaussian statistics. A number of different probability density functions have been taken into account by other researchers to describe, e.g., speckle statistics of biological tissues in optical coherence tomography, see e.g. [Ge, Rolland, Parker, Biomedical Optics Express 12, 4179 (2021)]. Also the Weibull distribution has been investigated in connection with light scattering by biological tissues, e.g. [Fawzym Zeng, Applied Optics 45, 3902 (2006)]. There is a huge literature concerning non-Rayleigh statistics in biological tissues or turbid media, but there is no mention of any of it in the manuscript and the work is not set in an appropriate context.

Response: We thank the Reviewer for raising this important aspect that we initially overlooked. The Reviewer's remark about non-Rayleigh speckle statistics in biological tissues is totally correct but, crucially, our RWs are not speckles but nonlinear modes. As discussed below, we are not observing non-Rayleigh speckles in laser transmission, which would be a linear and global property of the field, but the formation of localized wave packets with extreme intensity.

The phenomenon we report is thus profoundly different from the long-tail intensity distributions reported for biological tissues in optical coherence tomography (*Ge et al, Biomedical Optics Express 12, 4179, 2021*). These PDFs are indeed related to the complex shape of the biological scatterers and the multi-scale nature of the tissue structure.

In our case, two main facts indicate we are observing a novel phenomenon: (i) the dependence on the input power of both the intensity statistics and output pattern (Fig. 2) and (ii) a RW typical spatial size that significantly differs from the speckle size (now in Fig. 4). To our knowledge, none of these facts can be understood using only a linear wave scattering model from non-spherical scatterers. The analysis based on the Euler Characteristics that we add in the new Fig. 3 (please see also the response to Comment 1.1 of Reviewer #1) demonstrates that the deviations from Gaussian statistics are associated with the local geometry of the wave field, i.e., to the wave profile. Therefore, the formation of nonlinear structures gives the correct interpretation for the observed statistics.

Revision: The points have been clarified in various points of the manuscript, also adding the suggested references [change (9)].

Comment 2.2: Usually, rogue waves are defined by a certain amplitude exceeding a significant wave height of the system. There is no corresponding value mentioned in the manuscript. The authors only refer to extreme or anomalous high intensities, but they draw a difference between bright spots arising from random Gaussian statistics and spots observed in the heavy tail statistics. A much clearer definition and justification of rogue waves in the systems would be helpful.

Response: We thank the Reviewer for the useful suggestion. We now adopt the definition of rogue waves (RWs) derived from oceanography, following most of the literature on the topic (for a comprehensive review see *Akhmediev et al, J. Opt. 18 063001, 2016*). The oceanographic criterion identifies RWs as waves whose trough-to-crest height exceeds twice the significant wave height, the mean height of the highest third of waves. In optics, where generally the wave intensity is recorded, this definition becomes $I_{RW} > 2I_s$, where the significant intensity I_s is the mean intensity of the highest third of events. Since this criterion lacks a rigorous theoretical justification, there is also a diffuse consensus in referring to RWs when the intensity data presents long-tail statistics deviating from the expected exponential distribution. In the revised paper, we apply both criteria to identify bright spots of anomalous intensity as RWs. The statistics of the intensity maxima allows a more accurate identification of the single RW.

Revision: The RW definition has been clarified and the intensity threshold I_{RW} is reported in Fig. 2 and in the new Fig. 3 [change (10)].

Comment 2.3: From the point view of statistical appearance, bright spots with the same extreme and anomalous high intensities can also appear in the case of Gaussian statistics, which are even more rare but should still be observable. But those spots would be of a different type and would also exhibit completely different characteristics, as one has to conclude from Fig.3.

Obviously, a transmission of light with high intensities is possible in different ways. This raises a lot of questions. For example, what is the other physical mechanism for the transmission? Is only one type observed in the two different scenarios?

Response: Consistently with the above RW definition, NWs refer to bright spots in a wave field obeying to Gaussian statistics. As correctly noted by the Reviewer, events of extreme intensity can appear in both cases, but their features will be completely different (as we observe for their spatial size, for example) and sustained by a different transmission mechanism. To clarify the transmission type in the NW scenario, we study in detail the tumor spheroid scattering properties at low power (1 mW). Various evidence indicates diffusive scattering. First, the measured total reflectance is approximately 0.3, which in tissues indicates a ratio between the reduced scattering coefficient and the absorption coefficient larger than 10, a commonly accepted criterion for modeling light propagation by diffusion theory (see *Jacques et al, Journal of Biomedical Optics 13, 041302, 2008*). The absorption coefficient of the bio-printed samples without culture solution (0.25 cm^{-1}) is comparable with pancreatic tissues that are strongly diffusive (see *Lanka et al, Scientific Reports 12, 14300, 2022*, for optical characterization of pancreatic tissues). The Rayleigh distribution is hence the expected statistics, and its observation both for low-power and weak-coupling conditions is an additional robust indication of the diffusive regime when nonlinear effects are absent.

Revision: The points have been discussed in the manuscript [change (9)].

Comment 2.3: Why do the authors need the concept of rogue waves at all? The authors basically demonstrate that the system can easily be controlled, in the statistics as well as in a direct control of the event itself. This appears to be in contrast with the unpredictability criterion of the rogue wave phenomenon. Especially, I don't see any gain in deeper insight concerning the rogue wave phenomenon.

Response: The RW concept is the most appropriate to understand disordered wave fields presenting nonlinear effects, and this is exactly the case in our experiment where light transmission in tumors exhibits spots with extreme intensity and power-dependent behavior. The connection between these nonlinear filaments and the transmission spectrum is an important insight revealed by our study. This link, which to our knowledge has never been reported before, can be useful to bridge RWs in linear and nonlinear systems and may be relevant for controlling RWs in contexts of practical interest.

About the RW control we achieve, we do not share the Reviewer's point. Considering the complexity of our system, where linear and nonlinear optical effects interplays with biophysical processes, the fact that we succeed in switching on and off both the long-tail statistics and the appearance of the single RWs is a remarkable result. The unpredictability of a system does not mean that a certain degree of control is not achievable. In the context of our experiment, it implies that there is not a simple relation between the set of input modes and the RW properties. This is evident in Fig. 6, where the RW appears and disappears in a random manner for different phase masks.

Comment 2.4: How has the spatial intensity distribution for normal waves in Fig. 1 been received? Does it correspond to one realization observed in the investigation or has it been generated as a special case? This question concerns also the data which has been taken into account for the probability density function presented in Fig. 2 a. Does it contain the intensities of all measurements for one sample, or only for one random input wavevector spectrum $\{k_j\}$? This question is also related to my question above concerning the two scenarios.

Response and revision: All these points are clear in the revised manuscript [change (11)]. The statistical distributions in Fig. 2a, Supplementary Fig. 3, etc., are shown as computed from raw data, without any post-processing, for the single 3DTM sample and refer to many input spectra (i.e., many different high-resolution intensity images). The error bars in Fig. 2b refer to fluctuations from sample to sample. All the experimental PDF are statistically robust and not a special case. To give an idea, the dataset in Fig. 2a has 25600000 points (640 images of 200x200 pixels). In this case, we count with a custom algorithm the RWs by using the oceanographic criterion and we found approximately 2700 RWs.

Comment 2.5: The authors argue that the rogue wave formation is dominated by linear wave phenomena. It is difficult to follow this line of thought based on the data in Fig.2c. There are different numbers of peaks with high intensities for different powers and a not clear distribution to the other eigenchannels. The dynamics appear more complex, and the conclusion drawn from one peak for the total incoming energy is not directly accessible. The identification as a linear theory is an extremely important point, which should be supported by more clear data.

Response: The Reviewer's remark is totally correct. We apologize for the unclear interpretation in the initial version of the paper. The way we mixed linear and nonlinear concepts resulted in an unclear explanation. We now remove the ambiguity and stress that the observed RWs are nonlinear modes. As discussed in response to Comment 1.2 and Comment 2.1, the nonlinear wave scenario is fully supported by (i) the dependence of power of the statistics (Fig. 2b), (ii) the complex spatial dynamics of the output field (Fig. 2c), (iii) the difference in the RW size (Fig. 4), and (iv) the EC analysis (Fig. 3b). Note that the analysis in terms of the transmission spectrum remains valid. However, the transmission channels should not be interpreted as the eigenvalues of a linear system but as general spatial modes that can be excited by a specific shape of the input wave. In Fig. 6 we illustrate a direct application of this method.

Revision: We added new data in Fig. 4b [change (2)] that support the identification of the extreme waves with nonlinear modes. The interpretation of the observations has been clarified throughout the paper [change (1)].

Comment 2.7: The difference of spatial properties for weak and strong coupling is interpreted as resulting from a reduced spectral content for the weak coupling. Again, this is not obvious by the provided data. How has this been investigated? The authors emphasize on the light-induced thermal effects. This adds a further parameter into the coupling principle. Can the authors exclude the generation of rogue waves for other input powers for the weak coupling case?

Response: Strong and weak coupling refer to the distance between the 3DTM and the optical focal plane. The input wavevector spectrum is determined by phase modulation, thus the set of input wavevectors does not change in the two cases. The key difference is the typical size of the spatial modes impinging on the sample. When the spheroid is placed after the focal region, modes of a wider transverse size excite the 3DTM. RWs appear in this condition (strong coupling), while they are suppressed in the weak coupling case. This fact can be understood considering the shape and the larger spatial extension of the RWs (Fig. 4). A focused input beam with small coherence length (weak coupling) is unlikely to match the RW typical size, whereas a more extended wavefront with wider modes can form more easily localized filaments. This size-dependent behavior is a typical feature of spatial RWs of nonlinear origin (see Ref. 46).

Revision: The explanation of the strong and weak coupling condition has been clarified, and Fig. 4 has been corrected [change (12)].

Comment 2.8: The authors state that rogue waves in photonics form a noise-resistant tool for probing and manipulating the underlying material. This point is not clear. The authors should support this statement with a corresponding reference.

Response and revision: We agree. Unclear statements have been properly corrected as per previous changes.

REVIEWERS' COMMENTS

Reviewer #1 (Remarks to the Author):

The authors have convincingly clarified all the issues that I have raised. I'm happy with the revised version of the manuscript.

Reviewer #2 (Remarks to the Author):

The authors have addressed all my points and have revised the manuscript accordingly. Especially, they have clarified the nonlinear nature of the rogue waves observed in their system. They have extended their analysis and give a deeper insight into the observed phenomenon. All results are now placed on a more profound basis. The authors also added further substantial material related to a direct application in photo-thermal therapy. I can recommend the manuscript for publication in Nature Communications.

Point-by-point response to Reviewer #1

Report 1: The authors have convincingly clarified all the issues that I have raised. I'm happy with the revised version of the manuscript.

Response: We thank again the Reviewer for the very constructive review.

Point-by-point response to Reviewer #2

Report 2: The authors have addressed all my points and have revised the manuscript accordingly. Especially, they have clarified the nonlinear nature of the rogue waves observed in their system. They have extended their analysis and give a deeper insight into the observed phenomenon. All results are now placed on a more profound basis. The authors also added further substantial material related to a direct application in photo-thermal therapy. I can recommend the manuscript for publication in Nature Communications.

Response: We are pleased that the Reviewer finds the revised manuscript convincing and recommends publication. We thank again the Reviewer for the productive report that led us to the revised paper.